# A Modular Framework for Data Processing at the Edge: Design and Implementation

**DOI:** 10.3390/s23177662

**Published:** 2023-09-04

**Authors:** Lubomir Urblik, Erik Kajati, Peter Papcun, Iveta Zolotova

**Affiliations:** Department of Cybernetics and Artificial Intelligence, Faculty of EE & Informatics, Technical University of Kosice, 042 00 Kosice, Slovakia; erik.kajati@tuke.sk (E.K.); peter.papcun@tuke.sk (P.P.)

**Keywords:** containerization, edge computing, data processing framework, Kubernetes, Docker

## Abstract

There is a rapid increase in the number of edge devices in IoT solutions, generating vast amounts of data that need to be processed and analyzed efficiently. Traditional cloud-based architectures can face latency, bandwidth, and privacy challenges when dealing with this data flood. There is currently no unified approach to the creation of edge computing solutions. This work addresses this problem by exploring containerization for data processing solutions at the network’s edge. The current approach involves creating a specialized application compatible with the device used. Another approach involves using containerization for deployment and monitoring. The heterogeneity of edge environments would greatly benefit from a universal modular platform. Our proposed edge computing-based framework implements a streaming extract, transform, and load pipeline for data processing and analysis using ZeroMQ as the communication backbone and containerization for scalable deployment. Results demonstrate the effectiveness of the proposed framework, making it suitable for time-sensitive IoT applications.

## 1. Introduction

The proliferation of devices at the edge of the network, year-on-year increments in computing power, more energy-saving devices, and small form-factor devices are creating new kinds of technological challenges and are generating a significant volume of data that were not anticipated when the cloud-based computing paradigm was developed. The volume of data being processed, the prioritization of processes, and the requirements for a low response critical for some applications have led to shifting computing resources as close as possible to the sources of these data. Edge devices may be consuming as well as producing data, so it is necessary to move some aspects of the infrastructure closer. Edge computing and cloud computing are not mutually exclusive. The issue is about extending and offloading demand from remote servers and reducing the load on the global network [1].

The Internet of Things (IoT) plays a significant role in this computing paradigm. Efficient data processing has become a critical aspect of IoT applications, enabling better monitoring, analysis, decision-making, and automation of various applications [2]. However, efficiently processing and managing vast amounts of data poses significant challenges, particularly regarding latency, bandwidth, and privacy. Edge computing is an emerging paradigm that aims to address these challenges by processing data closer to their source, reducing the need for data to travel long distances to centralized data centers [3]. This approach results in lower latency, reduced bandwidth and consumption, and improved data protection. However, efficiently deploying and managing applications at the edge remains a complex task [4].

Containerization has proven to be a powerful technology for deploying and managing applications. It offers improved scalability, portability, and resource utilization. Containers enable lightweight, isolated environments for running applications, making it easier to manage and scale applications across heterogeneous edge computing infrastructures [5,6,7].

In addition, Infrastructure as Code (IaC) allows for the streamlined management of infrastructure resources, enabling consistent and repeatable deployments. Using IaC, developers and operators can automate the provisioning and management of infrastructure resources, reducing the likelihood of human error and increasing the efficiency of the deployment process [8].

The rapid development of dedicated edge devices brings new opportunities as the performance of the devices increases. The variety in architectures, platforms, performance, and power of these devices presents a challenge in ensuring the compatibility of created solutions. The lack of standardized communication between parts of the solutions also poses a problem. To ensure the reusability of created solutions and compatibility with as many devices as possible, we propose a novel framework based on Docker containers. The framework employs a pipeline approach to data processing while providing an easy way to modify the pipeline steps.

This work describes the implementation of the framework using a few custom-built services. By employing the pipeline approach, the framework allows for easy modification and extension of the services offered, providing a foundation for other solutions.

## 2. Background

This section provides an overview of the key concepts and technologies underpinning the proposed system, laying the foundation for a deeper understanding of the subsequent sections.

### 2.1. Containerization and Its Benefits

Containerization is a lightweight virtualization technique that runs applications in isolated environments. A container image packages an application with all its dependencies, allowing it to be executed consistently across different platforms and environments. Docker is one of the most widely used containerization platforms, providing a robust ecosystem of tools and services for building, deploying, and managing containers. As shown in Figure 1, many required dependencies are omitted in containerized applications [9].

### 2.2. Container Orchestration

Container orchestration is about managing and automating multiple containerized applications running on a cluster of machines instead of containerization, which focuses on creating isolated environments for applications and their dependencies. While containerization enables applications to run consistently across different platforms and environments, container orchestration ensures that these applications are efficiently deployed, scaled, and managed to meet high availability, resilience, and load-balancing demands. Kubernetes is an open-source container orchestration platform that simplifies the management of containerized applications across clusters of machines. Kubernetes provides various features, such as automatic scaling, rolling updates, and self-healing, to ensure that applications remain highly available and resilient [10].

### 2.3. Edge Computing and Its Importance

Rather than relying solely on centralized data centers or cloud infrastructure, edge computing is a distributed computing paradigm that moves data processing, storage, and analytics closer to the data source, such as IoT devices and sensors. This approach addresses several of the challenges associated with large-scale IoT deployments and offers multiple benefits:Reduced latency;Improved bandwidth utilization;Improved privacy;Improved scalability and efficiency.

Reduced latency is one of the key benefits of edge computing, as data are processed closer to their source, resulting in faster response times for applications that rely on real-time decisions. This is particularly important in scenarios such as autonomous vehicles, industrial automation, healthcare, and smart cities, where low-latency responses are critical to the safety, efficiency, and overall performance of the system [11].

Improved bandwidth utilization also comes into play as edge computing reduces the need to send raw data across the network to a central processing facility. By processing data locally, we can reduce network traffic and communication costs by minimizing the data transmitted over the network [12].

In addition, this approach improves privacy and security by allowing sensitive data to be processed and stored locally without leaving a local firewall. This helps protect data from potential breaches and leaks [13].

The system can handle increased data volume and workload better by distributing computing tasks across multiple edge nodes. Furthermore, by performing computations at the edge, data centers and cloud infrastructures can offload a portion of their workload, resulting in lower energy consumption and a reduced environmental impact [14].

### 2.4. Infrastructure as Code

Infrastructure as Code is an approach that automates the management and deployment of infrastructure using code and configuration files rather than through manual processes or custom scripts. IaC enables developers to define, version, and maintain infrastructure components consistently and repeatably, similar to how software applications are developed and maintained. The key benefits are consistency and repeatability, automation, and reduced costs. Depending on the specific tool, IaC can use declarative or imperative methods to create and manage the infrastructure [15].

### 2.5. Real-Time Data Processing Tools

Real-time data processing tools are crucial for modern applications that analyze and react to data in motion. These tools enable large volumes of data to be processed with low latency, ensuring that insights can be generated faster and decisions can be made quicker. In IoT and edge computing, real-time data processing tools enable rapid decision-making, anomaly detection, and dynamic system adaptation [16].

ETL processes are essential aspects of data processing. They involve extracting data from different sources, transforming it into a desired format, and loading it into a target system for further analysis or storage [17].

Streaming ETL is an extension of traditional ETL processes. It is specifically designed to handle continuous data streams in real time. Streaming ETL enables continuous data ingestion, transformation, and output with minimal latency. Unlike batch-based ETL, data are extracted, transformed, and loaded regularly. This approach is especially suited for modern applications with time-critical requirements like the IoT, event-driven architectures, and real-time analytics [18].

Streaming ETL processes have three primary components, as seen in Figure 2. First, data ingestion involves continuously consuming data from various sources, such as IoT devices or application events. Data ingestion components utilize streaming technologies like Apache Kafka or other messaging systems to receive and buffer data streams. Second, data transformation processes transform the ingested data in real time according to specified rules and logic. Transformation may include data cleansing, enrichment, aggregation, format conversions, or normalization. Lastly, data output involves loading transformed data immediately into target systems for further analysis, storage, or visualization. Output systems may include databases, data warehouses, or other analytics platforms, depending on the specific requirements of the application [19].

In this section, we will mention some works related to our article that served as either direct or indirect inspiration and solved similar problems or utilized the same technologies as our team. This section is divided into multiple subsections based on the main focus of the work.

### 2.6. Edge Computing

Al-Rakhami et al. [20] describe running regularized extreme learning machine neural networks in a containerized environment on an inexpensive edge device. The proposed framework divides the components into layers, which communicate using REST API to provide abstraction and independence to each component. The study focused on machine learning applications in an edge environment, specifically the classification of a person’s movement. The subjects were wearing multiple accelerometers and gyroscopes, the data from which were sent to a Raspberry Pi, which then processed and classified these data into five movement categories.

Kristiani et al. [21] created an air quality monitoring system using Docker, Kubernetes, and OpenStack. The devices collected data from the sensors and used MQTT to send them to the edge, where it was processed and saved, and in the case of an abnormal value, an alarm was sent using MQTT. The data were then sent to the cloud for long-term storage, analysis, and visualization. Docker created a unified environment for all edge devices and made deployment easier as the services were containerized and deployed across all devices.

Ren et al. [22] describe the advantages of edge computing with a focus on personal computing services. Their experiments compare various communication methods, namely Wi-Fi, BLE, 4G, and wired Internet, and their combinations. The tests are divided into three categories: edge only, edge and cloud, and cloud only. Their results show a decrease in latency whenever edge computing is used, especially whenever machine learning is applied, as the processing time is much shorter than the data transmission time.

González et al. [23] describe a data analysis pipeline used for biomedical images. The raw images are taken from imaging devices, such as microscopes or cameras, and with the application of AI, cells are detected and selected. Afterward, each cell is analyzed, the results of which are aggregated and saved. This pipeline approach gives the researchers more time for other tasks, such as experiments or research, as the data processing is almost entirely automated.

Abdellatif et al. [24] describe the various uses of edge computing in smart health. These include the detection of emergencies, such as falls, by using cameras or accelerometers situated in the room or on the patient. Another is a patient data aggregator, which uses various sensors spread across the patient’s body to measure vital signs. These are then sent to a nearby hub for processing and storage. Many heart-related emergencies are detectable by sensors long before the patient feels them, so latency becomes an important factor, and having a nearby edge node improves the quality of services that come with a direct-to-cloud connection.

Khan et al. [25] surveyed the various use cases of edge computing in smart cities. The expected rise of autonomous vehicles, which require a lot of processing power, can lead to quicker and better responses to traffic accidents. By placing edge nodes near the roads, they can contact the emergency units if a nearby vehicle has an accident. They can also evaluate the accident’s seriousness and appropriately relay the information to the rescuers [26]. Another possible use case is the detection of forest fires using unmanned aerial vehicles, such as drones. By either placing an edge node nearby or directly on the drone, we can ensure a much higher quality of service and, therefore, faster and better detection of possible fires. The UAVs can also serve as a communication infrastructure for rescue units in places where regular infrastructure is not sufficient [27,28]. The current parking infrastructure suffers from inefficient management, as drivers often have to drive all over and look for an empty spot. Some parking garages include distance sensors to detect where a car is present in a space, but these can be tricked with a shopping cart or cannot detect a parked motorcycle. By employing various artificial intelligence methods, such as neural networks, we can detect vehicles and their license plates and navigate the drivers to the nearest empty parking space [29,30].

Feng et al. [31] mention several possible edge computing applications in smart grids. As the voltage and frequency of electricity change dynamically, regulating these parameters requires real-time monitoring, which is a good fit for edge computing. Maintainers can also use these edge nodes, as they can provide accurate and real-time information about the current state of the infrastructure and help locate faults. Analyzing the data collected by the sensors connected to the distribution network can help owners forecast load or demand better. Using only the cloud is not feasible in such a case, as the amount of data collected would overwhelm the network and require less granularity, possibly resulting in a less accurate model. By collecting these data at the edge and then sending the results of computations to the cloud, the granularity is preserved, and higher accuracy can be achieved.

Meani et al. [32] proposed the utilization of edge computing in smart retail. By collecting data about users, such as demographics, and time-related data, such as time of year, time of day, and time spent in specific areas of the shopping center, in combination with center-related data, such as layout, paths, areas of interest, and the mapping of the stores, an application can offer personalized offers targeted at a single user. As shopping centers provide free Wi-Fi to users, communication between their mobile devices and the edge components occurs faster than with the cloud. This increases data throughput at a lower latency, leading to faster total processing time. Smart farming is another area that benefits significantly from adopting edge computing. Connecting the devices to the Internet might be problematic on bigger fields, as the signal might not reach them. Placing a few devices along the field that connect can help with this connectivity, but it will also lead to an increase in latency as the number of hops required to reach the cloud will increase. This becomes a non-problem if the processing takes place on the devices themselves.

Gireest et al. [33] created one such example: a Raspberry Pi is used with cheap, off-the-shelf sensors and actuators to create an automated irrigation system. As the temperature and humidity change, so must the irrigation. Otherwise, water might be wasted or insufficient. A more efficient system can be created by continuously measuring these values and adapting to the changes, leading to better yields.

Similarly to the detection of fires, drones can be used to monitor the fields, as described by Oghaz et al. [34]. By placing an edge computing module, such as an SBC, onto the drone, we can monitor the crops in real time and apply various detection algorithms. After a drone detects a diseased plant, it can move closer to the plant, pull it out, or spray it with a pesticide to prevent further spread. Moreover, if the drone is insufficient, it can mark the diseased plant for another robot or person to care for. Such monitoring can also monitor plant growth and predict yield and profit. Many nutrient deficiencies are also detectable from the air. Another possible use of UAVs in farming is the collection of data from the sensors, as these can be placed far out of reach. A drone regularly flies over these points and collects all the data without the devices having to send the data all the time.

One of the main limitations of edge computing is the power available to these devices. Typical cloud servers can consume hundreds or even thousands of watts of power. Pomsar et al. [35] describe the devices available for AI applications at the edge. Their study focused on devices with low power requirements: less than 40 W. The study also highlights another problem with edge computing: the variety of devices available. The performance of the devices in the study ranges from 0.472 tera operations per second (TOPS) to 32 TOPS, a difference of almost 7000% between the weakest and the strongest devices.

### 2.7. Frameworks

Pääkkönen et al. [36] propose a reference architecture for machine learning development and deployment in edge environments as an extension of the traditional Big Data reference architecture. Compared with a more traditional approach, which uses high-performance computers, edge environments may be constrained by size, performance, or energy consumption. This presents new challenges when developing machine learning solutions for edge, and requires the different processing and preprocessing tasks to be split between multiple environments and devices to achieve the best possible results. We consider this reference architecture an excellent starting point as it encompasses almost every part of edge computing. We find it lacks specific details on how the specific tasks and services should be created and deployed to the devices. This might lead to an incompatibility between different layers of the solution.

Bao et al. [37] propose a federated learning framework for use in edge-cloud collaborative computing, as the combination of federated learning and edge computing solves the privacy and security problems inherent in many areas, such as medical data. Computation offloading is widely used in the cloud-edge collaborative architecture, and the application of federated learning can take advantage of it to prolong the battery life of mobile devices. Another possible application is caching on the edge. To learn user preferences based on their age, gender, occupation, etc., the use of centralized learning becomes unavailable due to privacy concerns. By moving the learning to the edge, we can tailor the experiences to the users, as we can access their personal data without sharing it anywhere. This framework is focused on federated learning, which is an important part of edge computing but is not the only part. The framework could be extended to include other parts, such as data preprocessing, cleaning, or filtering.

Rong et al. [38] propose an industrial edge-cloud collaborative computing platform for building and deploying IoT applications. This platform utilizes a pipeline-based model for streaming data from IoT devices. The problem of heterogeneity in edge devices is solved by an abstraction that declares the properties and behaviors of devices. They also provide pre-implemented functions, which can be used as steps in the pipeline. The M:N relationship between devices and pipelines allows us to use multiple pipelines for a single device or connect multiple devices to a single pipeline. The effectiveness of this solution is described in a real-world example where multiple cameras were deployed to detect sewage dumping. The authors focused on AI for edge computing, more specifically, computer vision and the application of AI to video streams. We consider their pipeline approach an excellent tool, as it removes the need to redeploy the entire application in the case of minor changes. The authors make no mention of compatibility with various edge devices, which might prove problematic. They also train the models in the cloud, which sparks privacy concerns.

Lalanda et al. [39] created a modular platform aimed at smart homes that has since been expanded to other areas, such as smart manufacturing or smart building. This platform allows the connection of various devices, the development of data processing modules, and a dedicated simulator to test devices and modules. Unlike our framework, which is based on containerization and Docker, this solution is based on OSGi [40], a dynamic module system for Java. The modules can be created inside a dedicated IDE containing several wizards that help during the development. After creating a module, it can be deployed directly from the IDE to a remote platform. The framework displays great modularity but is limited to only one programming language: Java. We consider this a drawback, as many data-related and machine learning tasks are performed in Python. It can also lead to device incompatibility as it requires the Java Runtime.

Xu et al. [41] describe an edge computing platform that allows connected devices to communicate through HTTP or CoAP protocols. The platform allows bidirectional communication to obtain data from the devices or send commands, such as changing the parameters, to the devices. To connect a device, an object containing the basic information is created, containing values such as device name, protocol to be used, device address, port number, etc. When a device is connected, value descriptors must be set, which contain basic information about the value, such as type, min, max, and default value. This approach allows heterogeneous devices to be connected to a unified platform. The platform also allows for the creation of commands, which can contain parameters, expected return values, and descriptions. The abstraction of the connected devices provides a great way to ensure compatibility. The authors have focused only on the connection of devices and not the processing of the data exchanged between them. With some expansion and combination with a data processing framework, their method can serve as a great building block for edge computing solutions.

Trakadas et al. [42] propose a meta-OS, named RAMOS, aimed at edge computing. The authors consider the current hierarchical approach to the edge-fog-cloud continuum a barrier to the cooperation and collaboration of devices in IoT solutions. Their proposed solution, built on top of existing operating systems, aims to swap this approach for a more decentralized and distributed architecture. Unlike a more traditional approach in which cloud nodes take on the manager role, the authors propose multi-agent collaboration to achieve complex tasks without needing a central manager. By decentralizing the decision-making processes, the solution becomes more resilient to outages as it eliminates multiple points of failure. RAMOS aims to disrupt the current business models by returning the data back to producers. According to the authors, the current data balance between cloud and edge is 80–20%, highlighting the proliferation of cloud computing in the IoT. The authors envision a unified system, spanning everything from simple MCUs to high-performance servers, all using RAMOS. The nodes themselves are divided into two categories: atoms and molecules. Atoms are more simplistic nodes, providing basic computing capabilities and services. Molecules are more advanced nodes consisting of several Atoms and can provide the full functionality of RAMOS. These nodes advertise their capabilities, such as computing resources, services, and storage, using Agents. The Scheduler then looks at a list containing the Agents and decides where and how to process the data. RAMOS depends on a very high level of abstraction to unify the variety of devices used in the IoT. Implementing such an abstract solution may prove challenging due to different architectures, communication protocols and standards, and the sheer variety of available devices. Nevertheless, the potential of such a solution is immense and could transform the current approach to IoT solutions to be more collaborative, cooperative, and resource-efficient.

Srirama et al. [43] describe a fog computing framework named FogDEFT, which uses containerization to solve the problem of heterogeneity in IoT solutions. This framework aims to utilize the OASIS Topology and Orchestration Specification for Cloud Applications (TOSCA) modeling language in fog applications. Although traditionally aimed at cloud applications utilizing services from known providers, the authors were able to extend TOSCA to the fog. The services themselves use Docker to solve the problems of different operating systems. To solve the problem of different hardware platforms, the authors utilize Buildx, a tool provided by Docker that allows an image to be built for multiple platforms, i.e., AMD64 and ARM64. The framework was used for climate control in a convention center. The solution consisted of multiple Arduino MCUs, Raspberry Pi SBCs, and an AMD64 server.

One of the main advantages of cloud computing is access to virtually infinite resources. Recent years have seen a massive increase in the application of AI and ML in solutions. While powerful, these solutions tend to be demanding on the hardware. Edge devices cannot currently compete and require a slightly different approach. TinyML is a category of machine learning aimed at edge devices, which are constrained by the available performance, power, or size. The models created using this approach consume less energy and achieve better performance and comparable results when running on low-power devices such as SBCs. Lootus et al. [44] created a containerized framework for the deployment and monitoring of TinyML applications, named Runes, at the edge. One of the main drawbacks of AI at the edge is the need for the optimization of used hardware. As previously mentioned, the differences in available computing power in edge devices are immense. This extends to services available on these devices. The authors’ framework aims to optimize the created applications and ensure their compatibility with devices before deployment. The deployment of these applications is similar to Docker. The configuration is described in a Runefile, similar to a Dockerfile, which contains the basic information about the requirements of the image. It also contains the instructions that are executed when creating a Rune container. When deploying a new container, the framework first determines whether the device satisfies all requirements, such as communication protocols or connected devices. After a device is deemed capable, the container is created. To deploy Runes to multiple devices, the authors provide another tool named Hammer. This tool allows remote deployment to any connected device running RunicOS. We consider this work impressive, but the authors limit their framework to only AI/ML applications. There is no mention of any possible preprocessing to be performed using this framework, which limits the potential of this platform.

Edge computing provides an excellent opportunity for IoT solutions, as relying solely on the cloud may prove difficult in many situations. The works described in this chapter provide a great look at the potential of edge computing. However, this potential will not be fully utilized without a reference architecture. Due to the sheer number and variety of edge devices, a unified approach to this problem must be taken. During our research, we have found several key points that we consider vital to this problem:Containerization: By packing together all the required libraries and settings, we can remove the headache of finding a version compatible with our device.Container orchestration: Tools like Kubernetes allow us to remotely manage the containers running on our devices.Data format unification: The sheer number of IoT devices brings communication challenges. Different devices use different data formats to convey their data to the edge or cloud. By ensuring that the data are formatted before being processed, we can solve this problem.Modularity: Due to the wide array of IoT applications, the system should apply to as many of them as possible. An easy-to-modify framework that allows the developers to modify existing processing tasks and seamlessly add new ones will be necessary.

The table comparing the mentioned related works and our framework using the factors we consider important is shown in Table 1. The “x” denotes that the paper concerns itself with this category, the “-” that it does not.

## 3. Proposed Streaming ETL Framework

To address the challenges of real-time data processing in IoT applications, we present a comprehensive Streaming ETL framework. The design incorporates partially decentralized communication using the ZMQ event bus, an approach similar to [31], and MQTT broker, facilitating communication between the Streaming ETL services and the IoT devices. The modular system can be easily scaled thanks to a partially decentralized architecture.

Containerization technology, which provides a consistent environment for the application and its dependencies, plays a key role in ensuring the modularity and portability of the framework [41]. While simplifying deployment and management, this feature allows for smooth integration with infrastructure. As a result, the proposed framework can be easily adapted to different use cases and requirements, demonstrating its versatility in the face of diverse IoT application needs.

Furthermore, combining ZMQ and MQTT communication technologies enables efficient data transfer and processing even in high-volume or limited networked scenarios. The lightweight nature of MQTT makes it well-suited for constrained environments and low-bandwidth networks [38], while ZMQ’s asynchronous messaging capabilities provide reliable and high-performance communication between ETL services. Both chosen communication technologies use publish-subscribe architecture, contributing to the system’s modularity. The proposed architecture design with the event bus and event platform can be seen in Figure 3.

A robust and flexible infrastructure that can adapt to the dynamic demands of IoT applications can be achieved by combining Kubernetes and Terraform. With its declarative approach, Terraform enables seamless infrastructure management and versioning, paving the way for rapid development and deployment of the solution. Using Terraform with Kubernetes ensures that infrastructure changes can be applied consistently and reliably across environments, simplifying the transition from development to production [21].

This combination of technologies also promotes more manageable and maintainable infrastructure by encouraging the adoption of IaC practices. By treating infrastructure as code, the system’s configuration can be versioned and tested, increasing confidence in the stability of the deployed solution. Overall, the integration of Kubernetes and Terraform provides a solid foundation for the Streaming ETL framework, ensuring its adaptability, reliability, and efficiency in meeting the diverse needs of IoT applications.

Our programming language of choice for the Streaming ETL services’ development was Python. However, using the ZMQ event bus in our framework adds an extra layer of modularity, allowing the integration of components to be written in different programming languages such as C, C++, Java, JavaScript, Go, C#, and many others. This language-agnostic approach allows future researchers and developers to leverage the strengths of different programming languages when building individual components, increasing the flexibility and adaptability of the overall system.

As a result, the system can be easily extended by adding more services (subscribers and/or publishers) into the ETL ZMQ event bus, as can be seen in Figure 4, or can be customized to meet the unique requirements of different IoT applications and environments. By incorporating this level of modularity and versatility into the implementation, our framework becomes more robust and capable of handling the complex and evolving challenges associated with IoT data processing.

In the following sections, we will look at the details of each service we have implemented as part of our Streaming ETL framework. We will discuss their functionalities, used technologies, and how they work together to provide efficient real-time data processing tools for not only IoT applications. In this project, we have developed a standardized framework for real-time data processing that can be customized for different applications. By following this framework, the user can create a data processing pipeline tailored to their specific needs, taking advantage of containerization, edge computing, automated infrastructure on Kubernetes, and efficient communication protocols.

### 3.1. Input Data Transformation Service

The input data transformation service is an essential part of this framework. The data formats most commonly used in IoT solutions include JSON, raw values in either numerical or string form, and binary data. To ensure that all these types are usable in our framework, they must first be transformed into a unified format. The transformer processes raw data that are collected from various IoT devices. The service takes incoming data, applies transformation rules, and converts them into a standardized JSON format that can be easily consumed by subsequent components in the system, such as filtering and data analysis services. Data fields are standardized to general names during the transformation process, and measured values are rounded to two decimal places. After transformation, data are sent to the ZMQ event bus, making them available for other services. Standardization is the first step toward an autonomous AI algorithm capable of selecting and managing processing tasks to be applied to the data.

This service can receive data from MQTT devices via the MQTT broker, which we call push devices, or from REST API clients, which we call fetch devices. For REST API clients, we implemented a modular client that fetches data from the endpoint every *n* seconds, where *n* is configured in the environment variables of this service. By standardizing data, the input data transformation service increases the overall efficiency and interoperability of the system, ensuring a seamless integration with other back-end services.

### 3.2. Filtering Mechanism

Reducing the volume of data stored in the database is the responsibility of the filtering service. This is performed by applying pre-defined filters to the transformed data, ensuring that only relevant and necessary information is forwarded to subsequent components in the system.

By filtering data, the service helps optimize storage and processing requirements and reduce network bandwidth consumption. This efficiency is particularly important in IoT applications, where devices can generate massive amounts of data, but not all may be relevant or useful.

In addition, the service integrates modular filtering algorithms. Implemented filters include a value change filter that only sends data to the ZMQ event bus for further processing when a new value differs from the previous one. Another filter deals with numerical values with a set precision. In this case, data are only passed to the ZMQ event bus for further processing if a new value has changed by more than the set precision. These filters can be easily extended and adapted to suit different data processing requirements and scenarios.

### 3.3. Analyzing Service

Analyzing service is the next component in the Streaming ETL system, designed to perform analysis on transformed data to extract valuable insights. This service has three components: Redis Logger, Redis Consumer, and Redis Streams. These elements and dataflow can be seen in Figure 5.

Redis Logger reads transformed data from the ZMQ event bus and sends them to Redis Streams, an in-memory data structure for managing and processing real-time data streams. Redis Streams provides efficient stream processing capabilities, making it an ideal choice for IoT applications that require rapid analysis of large volumes of data.

Redis Consumer reads data from Redis Streams and analyzes the last *n* records. The analysis includes calculating the slope and rate of change of values such as temperature and humidity. These insights can then be used to support decision-making. It is important to state that within the analyzing service, any number of Redis consumers can be incorporated to meet the requirements of each specific application as needed. The service enables faster decision-making based on analyzed data by using Redis Streams for real-time data analysis.

### 3.4. MongoDB Time Series and Logging Service

MongoDB Time Series and logging service represent the final components of the Streaming ETL system, responsible for storing processed data from IoT devices. This service uses the MongoDB Time Series database, a specialized data storage solution designed to handle time-based data with high ingestion rates and large volumes. By utilizing a time series database, the system can efficiently store, index, and query large volumes of time-stamped data, making it easier to perform historical analysis and identify trends over time. In addition, the MongoDB Time Series provides flexibility to handle different data types commonly found in IoT applications, such as temperature, humidity, and many other sensor readings.

The logging service reads filtered data from the ZMQ event bus and stores them in the MongoDB Time Series database. In addition to transformed and filtered data, the logging service also includes information about the location of the device, the device ID, and metadata about the processes performed on the received data. This additional information provides valuable context for understanding and analyzing IoT data, enabling more informed decision-making and a better understanding of overall system performance.

### 3.5. Deploying Framework to Kubernetes Cluster

We deployed the containerized framework on a laboratory edge server running the Proxmox hypervisor. This is an open-source virtualization platform based on Debian GNU/Linux. Proxmox uses Kernel-based Virtual Machine (KVM) and Linux Containers (LXC) technologies to manage and create virtual machines and containers. We created two virtual machines that form a Kubernetes cluster consisting of a manager node and a worker node. The manager node is responsible for the management of the cluster and the coordination of its activities. In contrast, the worker node runs containerized ETL services, MongoDB Time Series database, and MQTT broker. Instead of using Dockershim to connect Kubernetes to the Docker container engine, we chose CRI-O, a Kubernetes Container Runtime Interface (CRI) implementation. This was performed because CRI-O is specifically designed and optimized to meet the requirements of Kubernetes. The full architecture of our framework is shown in Figure 6.

## 4. Testing of the Framework

To evaluate our framework, we have focused on analyzing the round-trip time (RTT) metric across ETL services within a series of tests. The tests included different numbers of measurements (10, 100, and 1000) and different time delays (1 ms, 10 ms, and 100 ms) between measurements. These parameters were selected to simulate different scenarios, such as a short burst with 10 values and 1 ms time delay or a long burst with 1000 measurements and 1 ms time delay. The burst, 1 ms time delay, can simulate an industrial sensor that requires an instant response to measured values. The longer delay can simulate commercial IoT sensors, such as temperature sensors, where the increased delay does not cause any problems. We expected the RTT to decrease as the time delay decreased due to the dynamic frequency boosting of modern CPUs. To test the framework, we have chosen two different approaches that allowed us to verify the efficiency and performance of our framework on different modern hardware platforms. First, we tested the framework on a 64-bit ARM processor and then on a 64-bit x86 processor. We allocated eight CPU cores and 8192 MB of RAM in both environments for the test environment.

The chosen architecture for testing the framework can be seen in Figure 7. Since we focused on testing ETL services within our proposed framework, we omitted actual IoT devices from our test environment because of the lack of computing resources and limited control over data transfer rates. For this reason, we designed a test container that could simulate an IoT device and allow us to adjust the speed and volume of the data feed as needed. The test container generates and sends data to the ZMQ event bus and then receives them to calculate RTT. We also omitted the MQTT broker and decided not to include the analysis service in the test as it does not play a primary role in data processing and depends on a required application. This proposed architecture allowed us to perform load tests on the designed services and ZMQ communication while giving us control over parameters we could change for different test scenarios.

### 4.1. The Test Environment

For our testing, we took two different approaches, which allowed us to compare the effectiveness and performance of two different hardware platforms. The first was a 64-bit ARM CPU, and the second was a 64-bit x86 CPU. This was performed to ensure compatibility of our framework with the two most common CPU architectures used in edge computing. We also wanted to test whether there were measurable differences in the performance of our framework between them. The test scenarios were also selected to see how they dealt with various workloads.

Table 2 shows the different platforms used in our testing. The Apple M1 CPU uses the ARM big.LITTLE architecture [45], employing high-performance cores (CPU-p), named Firestorm, and energy-efficient cores (CPU-e), named Icestorm. This architecture allowed a seamless switching of tasks between these different cores.

### 4.2. Tests on ARM64 with 100 ms Delay

Test No. 1 consisted of 10 measurements of RTT across ETL services with a 100 ms delay between measurements. The median of this set of measurements was 5.392 ms, and the average was 5.729 ms, implying that the measurements’ distribution was approximately symmetrical. The standard deviation was 1.265 ms, implying that most values were close to the average. Other details are shown in Table 3. From the plot in Figure 8, we conclude that there was no increased activity in efficient cores. The performance cores worked at high frequencies, regardless of the test.

Test No. 2 consisted of 100 measurements of RTT across ETL services with a 100 ms delay between measurements. The median was 5.313 ms, and the average was 5.055 ms, implying that most measurements were close to the average. The standard deviation was 1.965 ms, which means that the values of the measurements are significantly different from the average and thus the distribution of the measurements is quite scattered. Other details can be seen in Table 3. From the plot in Figure 9, we can see that in some periods of the test, the RTT time is significantly reduced due to the higher clock speed of the efficient cores. In these intervals, the efficient cores were at their maximum clock speed. The performance cores operate at high frequencies, regardless of the test.

Test No. 3 consisted of 1000 measurements of RTT with a delay of 100 ms between measurements, where the median was 5.208 ms, and the average was 4.932 ms, indicating that most measurements were close to the average. The standard deviation was 1.505 ms, indicating a slight scatter of measurements and confirming that the distribution of measurements was approximately normal. The other data can be seen in Table 3. The plot in Figure 10 did not show a significant improvement in RTT time during the higher clock speeds of the efficient cores over the longer measurement period, and the performance cores operated at high frequencies independently of the test.

### 4.3. Tests on ARM64 with 10 ms Delay

For the following tests, we have decided to omit the plots as they were too similar to the plots in previous tests and therefore had little value.

Test No. 4 consisted of 10 measurements of RTT with a delay of 10 ms between measurements. The median was 5.189 ms, and the average was 5.036 ms, indicating that most measurements were close to the average. The standard deviation of the set of measurements was 1.177 ms, indicating that the distribution of measurements was slightly scattered. The other data can be seen in Table 3. We found a lower median in test No. 4 compared to test No. 1, where the difference was 0.203 ms, indicating a smaller RTT metric.

Test No. 5 consisted of 100 measurements of RTT with a delay of 10 ms between measurements. The median was 4.692 ms, and the average was 4.492 ms, indicating that most measurements were close to the average. The standard deviation of the set of measurements was 1.175 ms, indicating that the values of the measurements were slightly different from the average. Thus, the distribution of the measurements is slightly scattered. The other data can be seen in Table 3. Compared to test No. 2, we observed a median time lower by 0.621 ms in test No. 5, which could indicate a lower RTT metric.

Test No. 6 consisted of 1000 measurements of RTT with a delay of 10 ms between measurements. The median was 4.561 ms, and the average was 4.218 ms, indicating that most measurements were close to the average. The standard deviation of the set of measurements was 1.449 ms, indicating that the values of the measurements were slightly different from the average. The other data can be seen in the table (Table 3). We observed a 0.647 ms lower median time in test No. 6 compared to test No. 3, which could indicate lower RTT metrics.

### 4.4. Tests on ARM64 with 1 ms Delay

Test No. 7 consisted of 10 measurements of RTT with a delay of 1 ms between measurements. The median was 4.352 ms, and the average was 4.653 ms, indicating that most measurements were close to the average. The standard deviation of the measurements was 0.837 ms, indicating that the measurements differed little from the average. Thus, the distribution of the measurements was poorly dispersed. The other data can be seen in Table 3. From the plot in Figure 11, we conclude that the efficient cores had no increased activity during the test. The performance cores were operating at high frequencies, regardless of the test. Compared to test No. 4, the median in test No. 7 was 0.837 ms lower, which could indicate better RTT metrics.

Test No. 8 consisted of 100 measurements of RTT with a delay of 1 ms between measurements. The median was 1.144 ms, and the mean was 1.852 ms, with a standard deviation of 1.617 ms, indicating a slight scatter in the measurements. The other data can be seen in Table 3. In the plot in Figure 12, we can see a significant decrease in RTT values at the maximum clock frequency of the efficient cores. The performance cores worked at high frequencies, regardless of the test. We observed a 3.548 ms lower median in test No. 8 compared to test No. 5, which could indicate a lower RTT metric.

Test No. 9 consisted of 1000 measurements of RTT with a delay of 1 ms between measurements. The median was 2.504 ms, and the average was 2.428 ms, indicating that most measurements were close to the average. The standard deviation was 1.013 ms, indicating that the measurement values are slightly different from the average, and thus the distribution of measurements is slightly scattered. The plot in Figure 13 shows the timescale improvement in RTT values during the maximum clock frequency of the efficient cores. The performance cores operated at high frequencies, regardless of the test. Compared to test No. 6, we observed a lower median in test No. 9 (difference of 2.057 ms). This may indicate lower RTT metric values. See Table 3 for more details.

### 4.5. Tests on AMD64 with 100 ms Delay

Test No. 1 consisted of 10 RTT measurements across ETL services with a delay of 100 ms between measurements. The median of the measurements was 2.616 ms, and the mean was 2.719 ms, indicating that the distribution of measurements was approximately symmetric. The standard deviation was 0.488 ms, indicating that most measurements were close to the average. Other data can be seen in Table 3. From the graph in Figure 14, it can be concluded that there was no significantly increased activity in the CPU cores during the test.

Test No. 2 consisted of 100 measurements of RTT across ETL services with a delay of 100 ms between measurements. The median of this set of measurements was 2.563 ms, and the mean was 2.571 ms, indicating that the distribution of measurements was approximately symmetric. The standard deviation was 0.239 ms, indicating that most measurements were close to the average. Other data can be seen in Table 3. From the graph in Figure 15, it can be concluded that there was an increase in the activity of the CPU cores during the test.

Test No. 3 consisted of 1000 measurements of RTT communication across ETL services with a delay of 100 ms between measurements. The median of this set of measurements was 2.529 ms, and the mean was 2.565 ms, indicating that the distribution of measurements was approximately symmetric. The standard deviation was 0.233 ms, indicating that most measurements were close to the average. Other data can be seen in Table 3. Test No. 3 showed increased activity of the cores during the test. The graph in Figure 16 shows that the clock speed of the cores increased at the beginning of the test.

### 4.6. Tests on AMD64 with 10 ms Delay

For the following tests, we decided to omit the plots as they were too similar to the plots in previous tests and therefore had little value.

Test No. 4 was performed with a delay of 10 ms between measurements and consisted of 10 measurements. The mean RTT was 2.162 ms, the median was 2.001 ms, and the standard deviation was 0.538 ms. Compared to test No. 1, where the delay was 100 ms, we can see that a lower delay between measurements resulted in lower RTTs. The other data can be seen in Table 3.

Test No. 5 consisted of 100 measurements of RTT communication across ETL services with a delay of 10 ms between measurements. The median of this set of measurements was 2.092 ms, and the mean was 2.170 ms, indicating that the distribution of measurements was approximately symmetric. The standard deviation was 0.391 ms, indicating that most measurements were close to the average. Other data can be seen in Table 3. Compared with the second test, we can see that a delay of 10 ms showed lower RTT values.

Test No. 6 consisted of 1000 measurements of RTT communication across ETL services with a delay of 10 ms between measurements. The median of this set of measurements was 1.955 ms, and the average was 1.991 ms, indicating that the distribution of measurements was approximately symmetric and most measurements were close to the average, as shown by the low standard deviation value of 0.231 ms. Other data can be seen in Table 3. Test No. 6 showed better results than Test No. 3. At a delay of 10 ms, the median RTT value was slightly better (1.955 ms versus 2.529 ms).

### 4.7. Tests on AMD64 with 1 ms Delay

Test No. 7 consisted of 10 measurements of RTT across ETL services with a delay of 1 ms between measurements. The median of this set of measurements was 1.582 ms, and the mean was 1.798 ms, indicating that the distribution of measurements was approximately symmetric. The standard deviation was 0.660 ms. Other data can be seen in Table 3. From the graph in Figure 17, it can be concluded that the activity of the CPU cores increased during the test. Test No. 7, with a delay of 1 ms, had a median RTT of 1.582 ms, while Test No. 4, with a delay of 10 ms, had a median RTT of 2.001 ms. This means that decreasing the interval between sending data also decreased RTT values.

Test No. 8 consisted of 100 measurements of RTT across ETL services with a delay of 1 ms between measurements. The median of this set of measurements was 1.395 ms, and the mean was 1.444 ms, indicating that the distribution of measurements was approximately symmetric. The standard deviation was 0.244 ms, indicating that most measurements were close to the average. The results of this test are similar to test No. 7, with a delay of 1 ms, which was conducted with fewer measurements. The other data can be seen in Table 3. The graph in Figure 18 shows a significant increase in the clock speed of the processor cores during the test. Comparing with test No. 5, we can see that a lower median RTT of 1.395 ms was achieved in test No. 8 compared to 2.092 ms in test No. 5.

Test No. 9 consisted of 1000 measurements of RTT communication across ETL services with a delay of 1 ms between measurements. The median of this set of measurements was 1.425 ms, and the mean was 1.475 ms, indicating that the distribution of measurements was approximately symmetric. The standard deviation was 0.317 ms, indicating that most measurements were close to the average. Other data can be seen in Table 3. Compared with previous tests, it can be seen that a short delay resulted in the lowest average value of RTT. Comparing tests No. 6 and No. 9, we can observe lower mean and median values in test No. 9. However, in test No. 9, the maximum value of RTT is significantly higher. In the graph in Figure 19, we can see an increase in the clock speed of the CPU cores during the test, which was fairly stable throughout the test.

## 5. Discussion

Our test results provide valuable insight into the performance and efficiency of ETL services across different hardware platforms and configurations. Using a simulated IoT device as a test container allowed us to focus on data processing speeds and communication between ETL services while maintaining control over transmission speed and other test parameters. In addition, the decision to test the framework on ARM64 and x86 64-bit processors allowed us to explore the performance and compatibility of our framework across different modern hardware architectures.

The results show that the proposed framework performs well under varying conditions, with the RTT metric remaining within acceptable limits throughout the tests. The tests also show that the framework can handle a range of data transfer speeds and measurement volumes, demonstrating its potential for scalability and adaptability to different IoT deployment scenarios.

However, it is important to note that, while the test environment is comprehensive, it does not cover all possible scenarios. Further testing with real IoT devices and different network conditions may reveal additional challenges and potential optimizations for the ETL services.

Figure 20 shows a significant difference in performance between the ARM64 and AMD64 processors in our evaluation of the RTT metrics. As the number of measurements increases, the ARM64 processor shows a greater decrease in median RTT than the AMD64 processor. This could be due to the differences in the architecture of the two processors. On the other hand, the AMD64 processor shows consistently low latency across all test scenarios, highlighting its suitability for real-time computing applications that require minimal latency. In addition, the data show that as the time delay between measurements decreases, the median RTT values for both processors tend to decrease, suggesting improved performance with more frequent requests. Overall, these results provide valuable insight into the performance of the two processors in real-time computing applications and can help select the appropriate hardware for specific use cases.

The last test on both architectures shows that even when reaching the highest CPU frequencies, the processing is not fast enough to process the data before new data are acquired. This is shown in the results, where the mean and the median increased. The processing queue will keep lengthening, and the processing times will increase.

The proposed framework shows promising performance and efficiency in processing and communication between ETL services. Further research and testing with real-world IoT devices, networks, and analytical services are required to better understand the framework’s potential for use in different IoT scenarios.

The testing was performed on both ARM64 and AMD64 CPUs, as SBCs do not use a unified CPU architecture, and we wanted to ensure that our framework was viable for both architectures. Further testing is required to check if the differences between architectures transfer to smaller devices.

Several research challenges still need to be addressed. The many devices used in edge environments make task offloading and load balancing difficult. For latency-sensitive tasks, the task offloading mechanism has to consider the distance to available nodes, their current load, and their potential performance. Load balancing is also a more complex task as the devices’ performance can vary greatly. Compatibility also needs to be considered, as the devices may not be capable of completing all the required tasks, and the system, therefore, has to consider the capabilities of other nodes [46].

Another research challenge arises in mobile edge computing, where the edge devices or the users connected to them move. The solution must predict the movement and ensure that the services are deployed to the devices when needed. The services also need to be deployed quickly. An example of this might be a shopping center with an unevenly spread-out crowd. As a large group of users moves, the solutions must scale the services up. It also needs to scale the services down when no longer needed, leaving room for other services to be deployed [47].

The security in edge computing also needs to be addressed, as the devices themselves may not support the security tools available for other devices. The variety of devices requires a particular approach to ensuring that all the devices are protected. Even a single vulnerable device endangers the whole network. The devices are also more prone to physical attacks where the attacker gains physical access to the devices. The data need to be protected on their entire journey from the sensor to the cloud, and they can be potentially intercepted at any step [48].

## 6. Conclusions

Edge computing is a fast-growing field hindered by a lack of standardization. The variety in edge devices collecting and generating the data proves to be an obstacle.

This article focused on creating a modular framework that allows developers to modify and create data processing tasks on the edge. We first looked at the tools that can help develop and deploy edge computing solutions like Docker, Kubernetes, and Terraform. We then described the parts of our data processing pipeline and the architecture of our framework that brought the pipeline together. Using containerization technology, the tasks can be easily deployed, scaled, and monitored using an orchestrator. The pipeline approach we have selected allows us to have better control over the processing performed. It also allows us to reuse existing parts in new pipelines, leading to shorter development times. We consider the following to be the main strengths of our framework: compatibility—using Docker, we can deploy our framework to a wide array of SBCs, Mini-PCs, or servers; modularity—new data processing tasks can be easily added to the processing pipeline; agnosticism—our framework is not tied to any programming language. Parts of the pipeline can use different languages or versions of the same language. We have also created a unified testing platform that can be used to evaluate and compare the performance of edge devices.

In our testing, we focused on the RTT metric in our pipeline. This was tested on both the ARM64 and AMD64 platforms, and both were compatible. We have found that increasing the frequency of sending data leads to decreased processing time. The testing was performed using a synthetic sensor capable of simulating different sensors and generating synthetic data. This tool will be expanded to include different scenarios and serve as a standardized tool for criteria evaluation.

In the future, we plan on extending the framework with a graphical user interface (GUI) similar to Apache Airflow. This approach will provide an easy way to modify the data processing pipeline and allow even non-programmers to use the framework in their solutions. We also plan on creating more modules that will be included with our framework and can be directly used or modified. These modules will include additional data analytics, anomaly detection, and machine learning methods. Including advanced monitoring tools, such as Grafana, is also planned. The user will have better and more user-friendly access to their data with tables and graphs. The extension of the framework to include cloud nodes is also in our scope, as we have previously mentioned that edge computing and cloud computing are not competing technologies but rather complement each other.

## Figures and Tables

**Figure 1 sensors-23-07662-f001:**
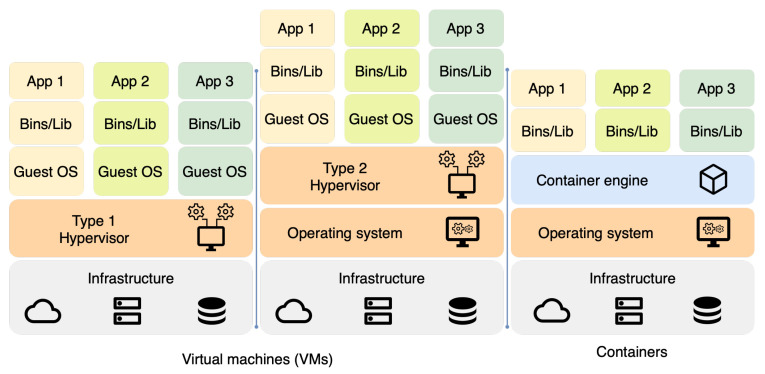
Architectures of virtual machines and containers.

**Figure 2 sensors-23-07662-f002:**
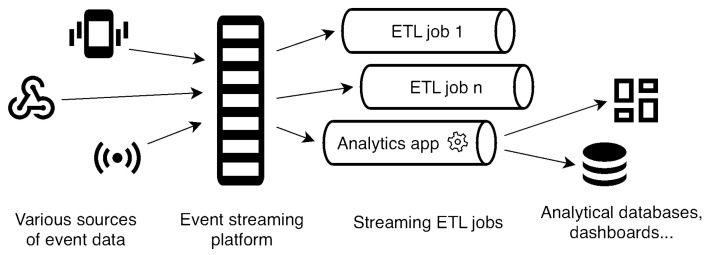
Streaming ETL data flow diagram.

**Figure 3 sensors-23-07662-f003:**
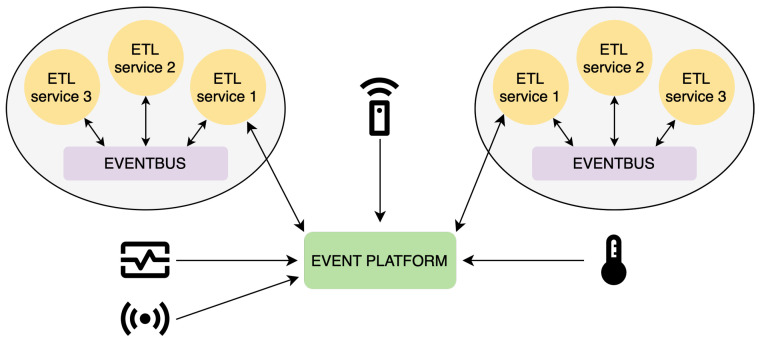
Architecture of the proposed framework.

**Figure 4 sensors-23-07662-f004:**
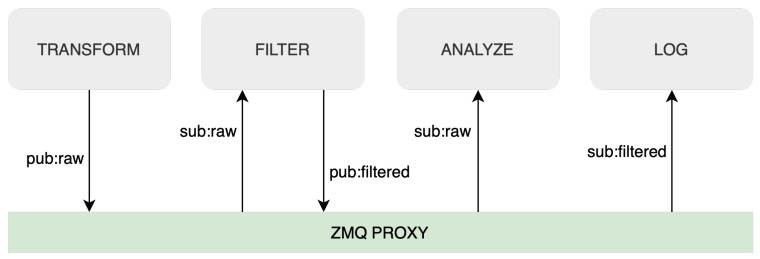
Streaming ETL ZMQ eventbus.

**Figure 5 sensors-23-07662-f005:**
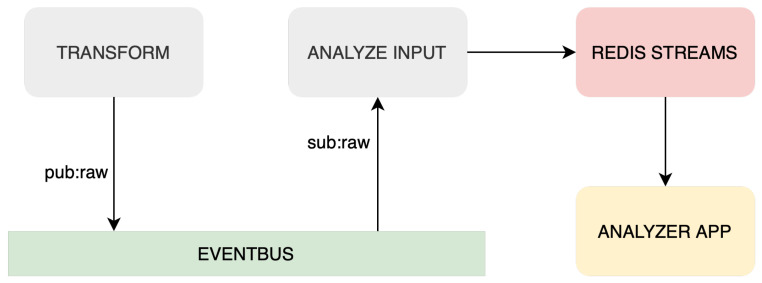
Dataflow of analysis service.

**Figure 6 sensors-23-07662-f006:**
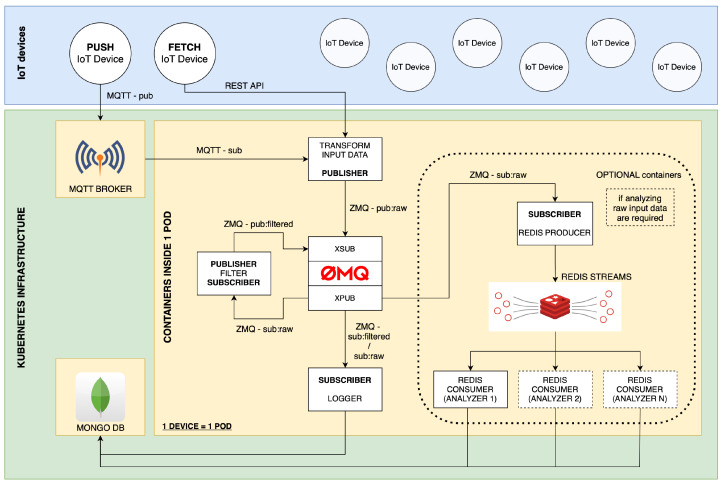
Full diagram of our framework.

**Figure 7 sensors-23-07662-f007:**
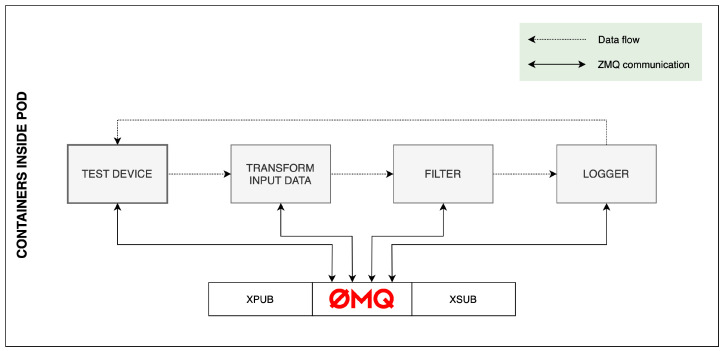
Communication between our containers.

**Figure 8 sensors-23-07662-f008:**
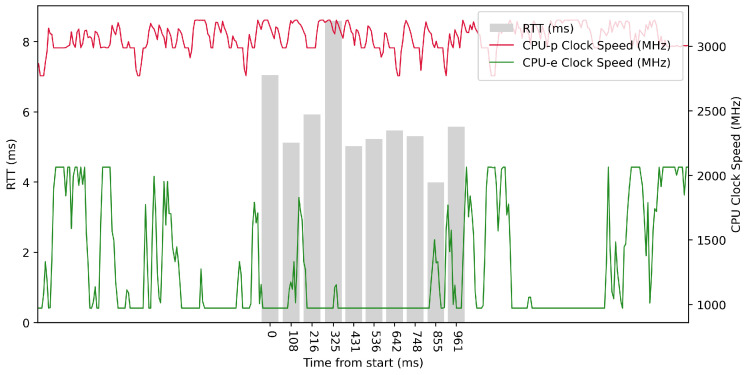
ARM64 Test No. 1: Round trip time across services, 10 times with a 100 ms delay.

**Figure 9 sensors-23-07662-f009:**
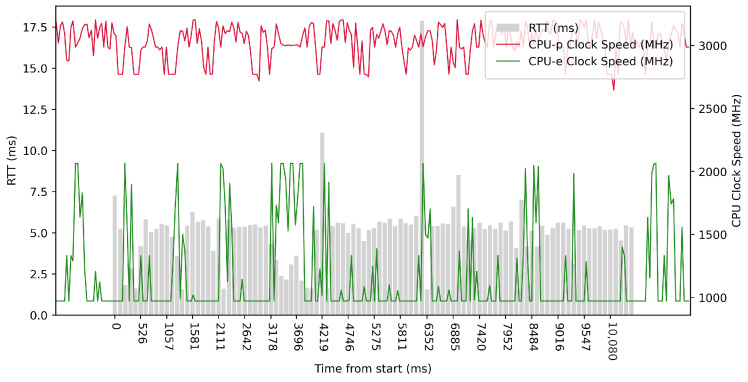
ARM64 Test No. 2: Round trip time across ETL services, 100 times with 100 ms delay.

**Figure 10 sensors-23-07662-f010:**
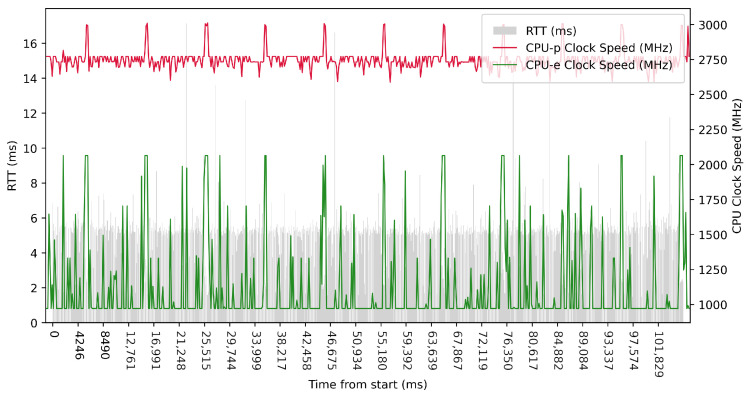
ARM64 Test No. 3: Round trip time across ETL services, 1000 times with 100 ms delay.

**Figure 11 sensors-23-07662-f011:**
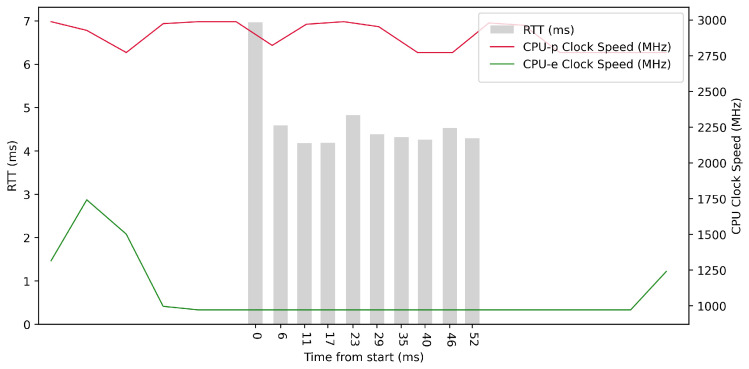
ARM64 Test No. 7: Round trip time across ETL services, 10 times with 1 ms delay.

**Figure 12 sensors-23-07662-f012:**
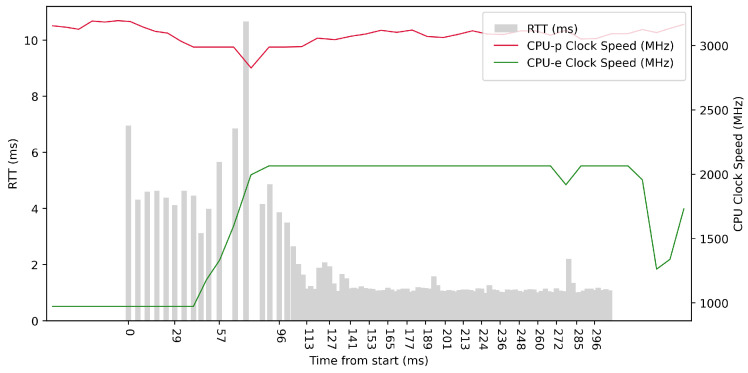
ARM64 Test No. 8: Round trip time across ETL services, 100 times with 1 ms delay.

**Figure 13 sensors-23-07662-f013:**
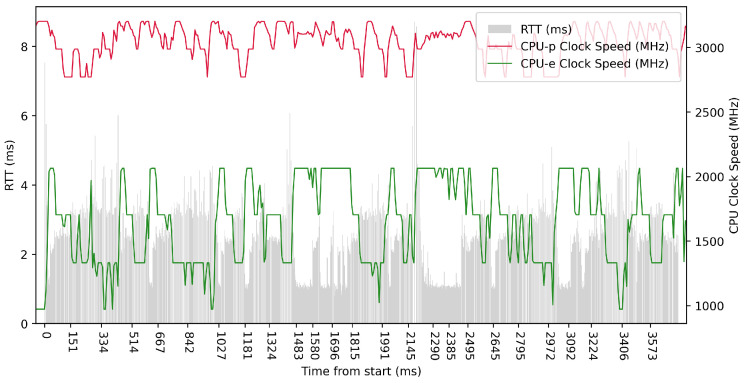
ARM64 Test No. 9: Round trip time across ETL services, 1000 times with 1 ms delay.

**Figure 14 sensors-23-07662-f014:**
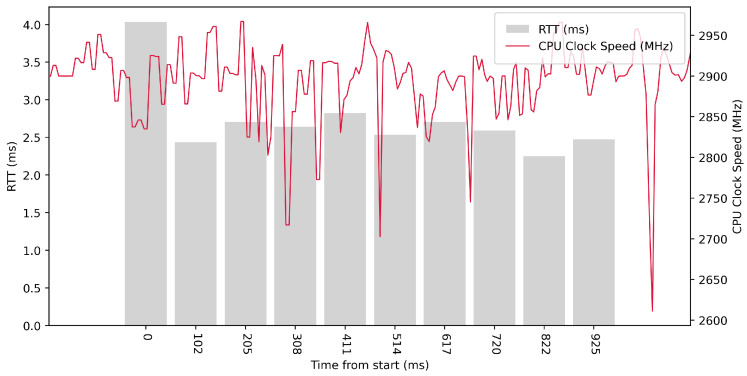
AMD64 Test No. 1: Round trip time across ETL services, 10 times with 100 ms delay.

**Figure 15 sensors-23-07662-f015:**
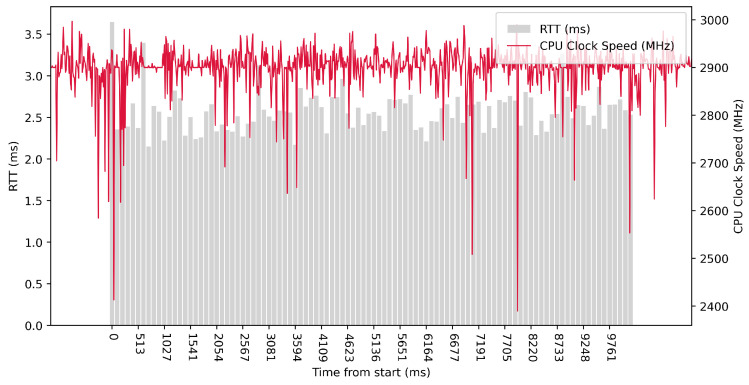
AMD64 Test No. 2: Round trip time across ETL services, 100 times with 100 ms delay.

**Figure 16 sensors-23-07662-f016:**
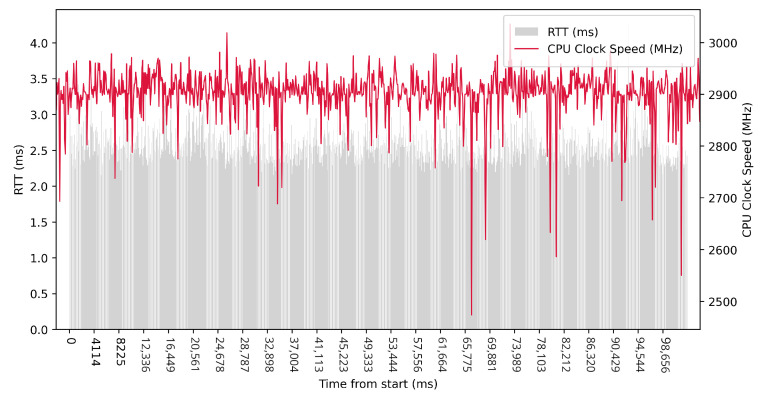
AMD64 Test No. 3: Round trip time across ETL services, 1000 times with 100 ms delay.

**Figure 17 sensors-23-07662-f017:**
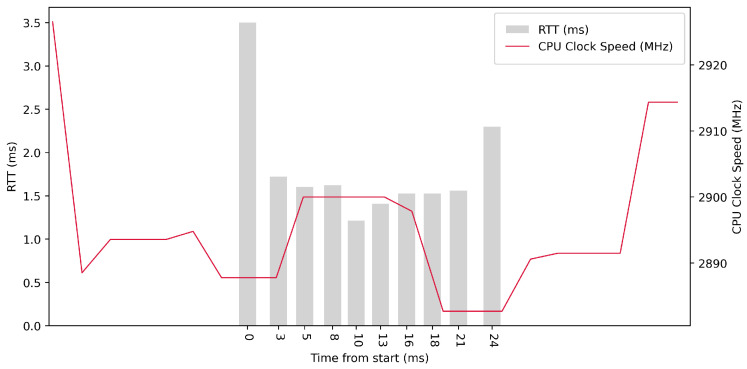
AMD64 Test No. 7: Round trip time across ETL services, 10 times with 1 ms delay.

**Figure 18 sensors-23-07662-f018:**
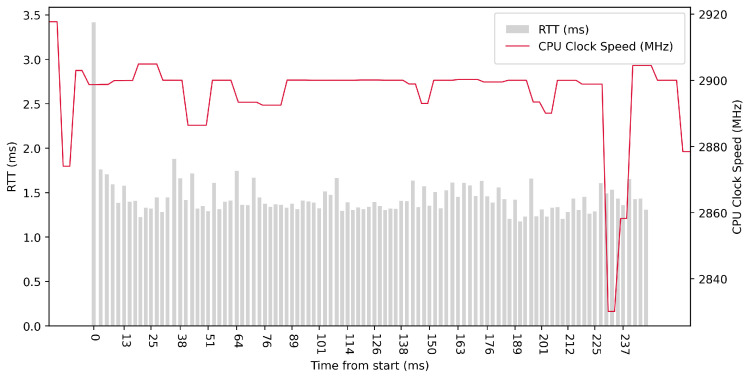
AMD64 Test No. 8: Round trip time across ETL services, 100 times with 1 ms delay.

**Figure 19 sensors-23-07662-f019:**
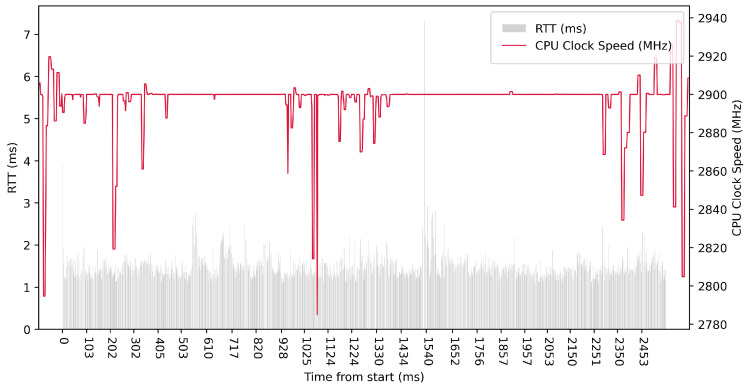
AMD64 Test No. 9: Round trip time across ETL services, 1000 times with 1 ms delay.

**Figure 20 sensors-23-07662-f020:**
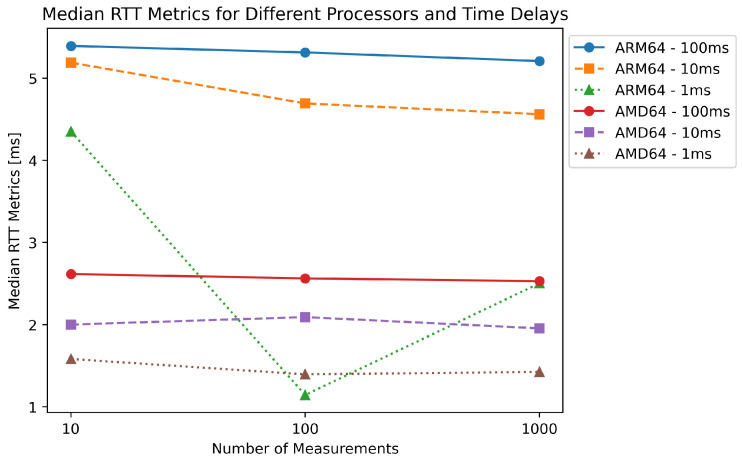
Median values for different architectures and delays.

**Table 1 sensors-23-07662-t001:** Comparison of the mentioned frameworks.

	Containerization	Orchestration	Data Preprocessing	Implementation
Pääkkönen, P. et al. [36]	-	-	x	x
Bao, G. et al. [37]	-	-	-	-
Rong, G. et al. [38]	x	-	-	x
Lalanda, P. et al. [39]	-	x	x	x
Xu, R. et al. [41]	x	x	x	x
Trakadas, P. et al. [42]	-	x	x	-
Srirama, Satish N. et al. [43]	x	x	x	x
Lootus, M. et al. [44]	x	x	-	x
Our framework	x	x	x	x

**Table 2 sensors-23-07662-t002:** Hardware used in testing.

	ARM64	AMD64
Device	Macbook Pro	HP Server
CPU	Apple M1	Intel Xeon Silver 4210R
CPU frequency	Firestorm: 600–3228 MHz Icestorm: 600–2064 MHz	2400–3200 MHz
CPU No. of cores	4× Firestorm (performance) 4× Icestorm (efficient)	12
RAM	16 GB	32 GB
Used cores	8	8
Used RAM	8 GB	8 GB

**Table 3 sensors-23-07662-t003:** The results of measurements.

CPU	Delay (ms)	No. of Measurements	Min (ms)	Max (ms)	Mean (ms)	Median (ms)	St. Dev. (ms)	Test Length (ms)
ARM64	100	10	3.989	8.594	5.729	5.392	1.265	961.095
		100	1.558	17.893	5.055	5.313	1.965	10,510.229
		1000	1.483	17.132	4.932	5.208	1.505	105,961.698
	10	10	2.916	7.528	5.036	5.189	1.177	139.874
		100	1.486	7.948	4.492	4.692	1.175	1536.522
		1000	1.258	16.844	4.218	4.561	1.449	15,173.540
	1	10	4.181	6.967	4.653	4.352	0.837	52.037
		100	0.995	10.667	1.852	**1.144**	1.617	306.143
		1000	**0.809**	8.707	2.428	2.504	1.013	3753.324
AMD64	100	10	2.249	4.031	2.719	2.616	0.488	925.488
		100	2.150	3.647	2.571	2.563	0.239	10,173.163
		1000	2.149	4.251	2.565	2.519	0.233	102,658.166
	10	10	1.844	3.620	2.162	2.001	0.528	109.515
		100	1.589	4.268	2.170	2.091	0.391	1216.439
		1000	1.564	3.531	1.991	1.955	**0.231**	12,107.133
	1	10	1.213	3.501	1.798	1.582	0.660	24.666
		100	1.176	**3.416**	**1.444**	1.395	0.244	248.040
		1000	1.007	7.309	1.475	1.425	0.317	2555.661

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
