# Peer review of "A Modular Framework for Data Processing at the Edge: Design and Implementation"

_sensors, 2023, doi:10.3390/s23177662_

Round 1
Reviewer 1 Report
Authors need to focus on the following issues:
1. Proposed work should be compared with State of Art techniques and there should be at least explanation for the same in abstract and it should be complemented in conclusion
2. Problem defination, motivation and contributions should be given at te end of introdcution. Different subsections should be formed for the same
3. Table is required for the related work and comparison with proposed work
4. Future work should be demonstrated in conclusion.
5. Research challenges are also required to discuss with applications.
Author Response
Dear reviewer, you can find answers to your comments in the points below. We tried to process all your comments and adjust them according to the instructions, so hopefully it will be fine. Our answers to comments are available below, also describing necessary changes which we made within the manuscript. The following changes that were made in the review article are displayed in the attached file.
- 1. Proposed work should be compared with State of Art techniques and there should be at least explanation for the same in abstract and it should be complemented in conclusion
- The abstract was changed to include the description of currently used techniques (specialized applications based on device used)
- 2. Problem defination, motivation and contributions should be given at te end of introdcution. Different subsections should be formed for the same
- The ending of the introduction chapter was modified to include the aforementioned categories.
- 3. Table is required for the related work and comparison with proposed work
- The table has been added, focusing on what we believe to be important parts of an edge framework
- 4. Future work should be demonstrated in conclusion.
- The conclusion was reworked to include the work we plan on doing in the future.
- 5. Research challenges are also required to discuss with applications.
- We have added three research challenges we consider to be important to our work and will need to be addressed in the future. They are available at the end of the "Discussion" section.

Reviewer 2 Report
This paper proposes a modular containerization framework based on Infrastructure as Code tools to deal with real-time IoT services. The performance evaluations are focused on the Round-Trip-Time metric under various measurements and delay configurations. Please consider the comments below in the revised version of this manuscript.
- The summary of the contributions of this paper is required.
- Possible limitations of each research work in the related work should have analyzed.
- Related work should have included newer related papers that are able to cover containerized solutions at the edge networks.
- What and how the “bandwidth and privacy challenges” have the authors indeed solved in this paper and then proven (statistics) by the proposed paradigm as mentioned throughout the paper?
- What are the weaknesses of the proposed framework?
- The authors omitted the IoT devices from the simulation environment, so how can the test container guarantee the actual performance of the real IoT devices and further the applicability of the proposed framework?
- The motivations and selections of several test configurations and the expected outcomes from those tests should be described in Section 5.
- In Fig. 26, why is the performance of ARM64-1ms (green color) suddenly changed? Is there any problem with simulated results?
- How can the authors guarantee the reliability of the simulation results?
- The framework should be publicly open-source, that can become the reference for the future work.
The quality of English language is fine.
Author Response
Dear reviewer, you can find answers to your comments in the points below. We tried to process all your comments and adjust them according to the instructions, so hopefully it will be fine. Our answers to comments are available below, also describing necessary changes which we made within the manuscript. The following changes that were made in the review article are displayed in the attached file.
- - The summary of the contributions of this paper is required.
- The "Conclusions" section was changed to include and better highlight the contributions of this paper.
- - Possible limitations of each research work in the related work should have analyzed.
- The related work section has been divided into two subsections - Edge computing and Frameworks. For each work in the Frameworks subsection, we extended the description to include what we consider to be good about the framework and what we consider to be limitations.
- - Related work should have included newer related papers that are able to cover containerized solutions at the edge networks.
- New papers, which utilize containerization, were added to the related works.
- - What and how the “bandwidth and privacy challenges” have the authors indeed solved in this paper and then proven (statistics) by the proposed paradigm as mentioned throughout the paper?
- As our framework is not yet finalized we have not addressed these challenges. We wanted to focus on the approach to data processing and communication between the services in our framework. These challenges will be addressed in the future with the creation of new services aimed at them.
- - What are the weaknesses of the proposed framework?
- The modular approach we have taken may prove a liability when it comes to latency, compared to a more tightly integrated solution. This remains to be seen and will have to be tested. The framework also requires the users to create their own modules and is therefore currently not recommended for non-programmers.
- - The authors omitted the IoT devices from the simulation environment, so how can the test container guarantee the actual performance of the real IoT devices and further the applicability of the proposed framework?
- The tests measured only the performance of the proposed framework, not the latency or quality of the communication. Our measurements started when the data was received by our framework, not when it was sent from the device. We wanted to focus our attention on the performance of the framework without external factors.
- - The motivations and selections of several test configurations and the expected outcomes from those tests should be described in Section 5.
- We have added an explanation behind our motivation and reasoning on the selected parameters and the expected outcome from them.
- - In Fig. 26, why is the performance of ARM64-1ms (green color) suddenly changed? Is there any problem with simulated results?
- We reran the tests multiple times and always achieved similar results. We believe it has to do with the load-balancing in Apple M1 processors and hitting some "sweet spot" where the CPU achieved highest performance without moving the workload between performance and efficiency cores.We plan on further testing this theory to see whether it can be achieved using different time delays and number of measurements
- - How can the authors guarantee the reliability of the simulation results?
- The tests were reran multiple times on multiple devices running the same hardware and with the same setup. The devices were running clean install OSs and were disconnected from the network to ensure the same conditions for all the devices used. The differences in achieved results were within a margin of error.
- - The framework should be publicly open-source, that can become the reference for the future work.
- We are planning on open-sourcing our solution, but we do not yet consider it to be sufficient as we are still testing it. We want to ensure that it can serve as a solid building block.

Reviewer 3 Report
Although "A Modular Framework for Data Processing at the Edge: Design and Implementation" is an interesting work focusing on edge computing, I have some minor comments prior publication:
1) There are some subsections that are unnecessary, such as the definition of containarized environments, bandwidth and latency constraints, etc.
2) The authors at the end of Section 2 simply state that "However, this potential will not be fully utilized without a reference architecture" without providing more details on the drawbacks of the studied references or to the key novelty of their work.
3) How many edge nodes do you use for the evaluation of your approach? How would you validate the scalability of any proposed solutions?
4) There is a huge amount of result figures for different categories of testing, and the reader is really lost. Perhaps a table summarizing the key findings per setup would be helpful.
5) Among other recent publications, the following paper can be considered to be added in the literature review: Panagiotis Trakadas, Xavi Masip-Bruin, Federico M Facca, et al., "A Reference Architecture for Cloud–Edge Meta-Operating Systems Enabling Cross-Domain, Data-Intensive, ML-Assisted Applications: Architectural Overview and Key Concepts", Sensors, vol. 22, issue 22, 2022.
No specific comments.
Author Response
Dear reviewer, you can find answers to your comments in the points below. We tried to process all your comments and adjust them according to the instructions, so hopefully it will be fine. Our answers to comments are available below, also describing necessary changes which we made within the manuscript. The following changes that were made in the review article are displayed in the Difference.pdf
- 1) There are some subsections that are unnecessary, such as the definition of containarized environments, bandwidth and latency constraints, etc.
- We have removed the division into subsections but left the texts in as we believe they can help new readers better understand our paper.
- 2) The authors at the end of Section 2 simply state that "However, this potential will not be fully utilized without a reference architecture" without providing more details on the drawbacks of the studied references or to the key novelty of their work.
- The comment was not aimed towards any of the mentioned related works but rather at the research area as a whole. Other areas, such as Industry 4.0 or IoT have reference architectures or models, which allows for easier development of new solutions or extensions of existing solutions. We believe that edge computing can greatly benefit from having a reference architecture
- 3) How many edge nodes do you use for the evaluation of your approach? How would you validate the scalability of any proposed solutions?
- We used only one node in each of our tests as we were focusing on the. We are planning on running multi-node tests in the near future to evaluate the scalability.
- 4) There is a huge amount of result figures for different categories of testing, and the reader is really lost. Perhaps a table summarizing the key findings per setup would be helpful.
- The number of figures was reduced from 18 to 12, making it easier to see the differences between 100ms and 1ms delays.
- 5) Among other recent publications, the following paper can be considered to be added in the literature review: Panagiotis Trakadas, Xavi Masip-Bruin, Federico M Facca, et al., "A Reference Architecture for Cloud–Edge Meta-Operating Systems Enabling Cross-Domain, Data-Intensive, ML-Assisted Applications: Architectural Overview and Key Concepts", Sensors, vol. 22, issue 22, 2022.
- The mentioned paper, along with some others, have been added to the related works.

Round 2
Reviewer 2 Report
Thanks the authors for their effort answering the reviewers' comments.
In general, all my comments have been resolved. Some are not indeed straightforward to answer, but I had known your problems. Additionally, the authors must submit the correct files (e.g., author_response.pdf) in the right place, and highlight where you made changes in the revised manuscript.